# *Lactobacillus rhamnosus* GG Reduces *β-conglycinin-*Allergy-Induced Apoptotic Cells by Regulating *Bacteroides* and Bile Secretion Pathway in Intestinal Contents of BALB/c Mice

**DOI:** 10.3390/nu13010055

**Published:** 2020-12-27

**Authors:** Xiaoxu Chen, Yuekun Wu, Yaozhong Hu, Yan Zhang, Shuo Wang

**Affiliations:** 1Key Laboratory of Food Nutrition and Safety, Ministry of Education, Tianjin University of Science and Technology, Tianjin 300457, China; chenxiaoxu_@126.com; 2Tianjin Key Laboratory of Food Science and Health, School of Medicine, Nankai University, Tianjin 300071, China; yuekunw@126.com (Y.W.); yzhu@nankai.edu.cn (Y.H.); wangshuo@nankai.edu.cn (S.W.)

**Keywords:** *β-conglycinin*, allergy, apoptotic cells, LGG, gut microbiota, metabolome

## Abstract

Allergy can cause intestinal damage, including through cell apoptosis. In this study, intestinal cell apoptosis was first observed in the β-conglycinin (β-CG) allergy model, and the effect of *Lactobacillus rhamnosus* GG (LGG) on reducing apoptosis of cells in the intestine and its underlying mechanisms were further investigated. Allergic mice received oral LGG daily, and intestinal tissue apoptotic cells, gut microbiota, and metabolites were evaluated six and nine days after intervention. Terminal deoxynucleotidyl transferase-mediated dUTP nick-end labeling (TUNEL) analysis revealed that LGG intervention could reduce the incidence of cell apoptosis more effectively than natural recovery (NR). The results of 16S rRNA analysis indicated that LGG intervention led to an increase in the relative abundance of *Bacteroides*. Metabolite analysis of intestinal contents indicated that histamine, *N*-acetylhistamine, *N*(α)-γ-glutamylhistamine, phenylalanine, tryptophan, arachidonic acid malate, and xanthine were significantly decreased, and deoxycholic acid, lithocholic acid were significantly increased after the LGG intervention on β-CG allergy; the decreases in histamine and *N*(α)-γ-glutamylhistamine were significant compared with those of NR. In conclusion, LGG reduces apoptosis of cells induced by β-CG allergy, which may be related to regulation of *Bacteroides* and the bile secretion pathway.

## 1. Introduction

β-Conglycinin (β-CG) is the main allergen of soybean. β-CG is a trimeric structure composed of α (67 kDa), α’ (71 kDa), and β (50 kDa) subunits. Each subunit is composed of an acidic peptide chain and a basic peptide chain, and the structure is very stable and therefore persistent, resulting in relatively constant sensitization toward β-CG [1,2]. Allergies can cause disorders of the immune system and cell apoptosis. Similarly, allergies are accompanied by the appearance of a large number of inflammatory cells whose survival is regulated by apoptosis. Hence, an increase in apoptotic cells helps to accelerate the inflammatory process [3,4]. Apoptotic cells may also damage intestinal mucosal immunity [5]. At present, the main challenge in managing allergies is related to the lack of effective measures and potential mechanisms for reducing apoptotic cells.

Probiotics can prevent human intestinal diseases. When appropriately consumed, they produce a beneficial effect on the host through immune regulation. Some studies have confirmed that probiotics have immunomodulatory effects on allergies [6,7]. Probiotics can change the composition of gut microbiota and regulate immune response of the host as well as the intestinal mucosa through effects on antigens [8,9,10]. The potential of probiotics to alleviate allergic response lies in their ability to affect processes involved in the repair of the intestinal barrier function, which regulates balance in the imbalanced microbiome and, thereby, the immune system [11].

Gut microbiota are essential for healthy immune regulation and gut barrier function. The gut microbiota can interact with intestinal metabolites, such as bile acids (BAs), and regulate the host metabolism mainly by activating immune genes in the small intestine to regulate microbial composition in the body [12,13]. BAs regulate immune and inflammatory processes through signaling transduction, such as through the farnesoid X receptor (FXR)-regulated pathway and cell surface G protein-induced signaling [14]. BAs can control the release of intestinal microbial immunoglobulin A (IgA) antibodies [15] and play an essential role in immune homeostasis of the intestine. BAs have anti-inflammatory effects on intestinal epithelial cells [16].

*Lactobacillus rhamnosus* GG (LGG) is one of the well-known commercialized probiotics. According to reports, LGG produces a preventive effect on the process of birch pollen allergic asthma [17]. LGG inhibits allergic inflammation in asthmatic mice [18] and alleviates allergic airway inflammation through gut microbiota [19]. LGG consumption results in a significant improvement in patients with milk protein allergy [20], and may alleviate allergy by improving intestinal homeostasis [21]. We previously verified that LGG can alleviate β-CG allergy by modulating immune gene expression in the T cell receptor (TCR) signaling pathway [22].

However, there is no published research on whether LGG can reduce cell apoptosis and whether the alleviating effect is related to the metabolism of the gut microbiota. In this work, the main purpose was to investigate the effect of LGG on β-CG-allergy-induced cell apoptosis as well as its related mechanism on the gut microbiota and related metabolic pathways.

## 2. Materials and Methods

### 2.1. Allergen

The separation and purification process of β-CG was performed based on a previously reported method [23].

### 2.2. LGG

The activation, culture and gavage concentration of LGG referred to our previously reported method [22].

### 2.3. Animals

BALB/c mice (4–6 weeks) were used in this experiment. The experimental conditions were based on our previously reported method [22]. Animals experiments procedures were approved by the Institutional Animal Ethics Committee (IRM-DWLL-2020094).

### 2.4. Experimental Design

The saline group and the control group were supplemented with saline (300 μL/mouse per week) and cholera toxin (CT; 300 μL/mouse per week), respectively. The β-CG group was supplemented with β-CG and CT for 5 weeks to induce allergy. Next, the allergic mice received oral LGG (1 × 10^9^ CFU/600 μL/mouse/day), had a normal diet, and were divided into four equal groups. The LGG-1 group mice and the LGG-2 group mice were supplemented with LGG and were sacrificed on day 6 and 9, respectively. The NR-1 and the NR-2 group mice received a normal diet and were sacrificed on day 6 and 9, respectively.

After all the experimental mice were sacrificed, different parts of intestinal tissue were separately fixed in 4% paraformaldehyde for terminal deoxynucleotidyl transferase-mediated dUTP nick-end labeling (TUNEL) analysis. The intestinal contents were obtained using a sterile cotton swab and immediately placed in a sterile cryotube until 16S rRNA sequencing analysis. The remaining intestinal contents were collected and stored for metabolomics analysis.

### 2.5. Allergy Symptom Score

After the fifth gavage, mice were observed for clinical symptoms, which were scored according to a previously reported method [22].

### 2.6. Analysis of Immunoglobulin E (IgE) and Histamine (HIS) in Mice Serum

The contents of IgE and HIS were determined by enzyme linked immunosorbent assay (ELISA), referring to the manufacturer’s instructions (Nanjing Jiancheng Bioengineering Institute, Nanjing, China).

### 2.7. Detection of Apoptotic Cells by TUNEL Assay

The intestinal tissue (4 μm) sections were deparaffined, rehydrated, and permeabilized with proteinase K. An In Situ Cell Death Detection Kit (Roche Diagnostics Ltd., Kanton Bern, Switzerland) was used for TUNEL detection based on detection of fluorescein isothiocyanate according to the manufacturer’s instructions, and apoptotic cells were observed. The total number of cells was counted after counterstaining of sections. The apoptotic cells were regarded as TUNEL-positive cells if they demonstrated obvious apoptotic morphology. Five fields of view were randomly selected to analyze the samples under a microscope with a 40× objective.

### 2.8. 16S rRNA Gene Sequencing Analysis

First, the DNA in the contents of the mouse intestine was extracted and sequenced using IonS5TMXL, and we conducted sequences analysis using Uparse software (Uparse v 7.0.1001, http://drive5.Com/uparse/). Taxonomic information of the representative sequence was annotated based on the Silva Database (https://www.arb-silva.de/) through the Mothur algorithm. Multiple sequence alignments were finalized by utilizing the MUSCLE software (Version 3.8.31, http://www.drive5.com/muscle/) to study phylogenetic relationship of different Operational Taxonomic Units (OTUs), as well as the difference of the dominant species in different groups. The data from OTUs clustering and species classification were analyzed. The related downstream analysis and data visualization were accomplished by QIIME (Version1.7.0) and displayed with R software (Version 2.15.3).

### 2.9. Metabolomics Analysis

Approximately 100 mg of mouse intestinal contents were ground separately with liquid nitrogen and resuspended in methanol and formic acid. The samples were incubated on ice and centrifuged, and we adjusted the final concentration to a solution containing 60% methanol. The raw data files generated after UHPLC-MS/MS were processed by using the Compound Discoverer 3.0 (CD 3.0, Thermo Fisher) to realize peak alignment, peak picking, and quantitation of each metabolite. Finally, the predicted the molecular formula was compared using the mzCloud (https://www.mzcloud.org/) and Chemspider (http://www.chemspider.com/) databases. KEGG pathway analysis was packaged based on Python software (Python-3.5.0).

### 2.10. Statistical Analysis

All data analyses were performed using SPSS 17.0 software (SPSS Inc, Chicago, IL, USA) and were expressed as mean ± standard deviation. One-way ANOVA nonparametric testing was performed to determine statistical significance. *p* < 0.05 was considered statistically significant.

## 3. Results

### 3.1. β-CG Induced Intestinal Cell Apoptosis in Allergic Mice

We first established the β-CG allergy model to observe the influence on intestinal apoptotic cells. The clinical symptoms of the mice administered oral saline and CT were normal. However, the allergic mice demonstrated significant clinical symptoms of allergy, such as diarrhea and erect hair (Figure 1A). IgE and HIS were significantly increased in the allergic mice (Figure 1B,C). Using the allergy symptom score and the typical allergy indicators IgE and HIS, we found that the β-CG allergy model was successfully established. The percentage of apoptotic cells in allergic mice was significantly higher than in the other groups (Figure 1D), indicating that β-CG allergy induced cell apoptosis. The pathological status of the duodenum, jejunum, ileum, and colon was determined using TUNEL assay (Figure 1E–H). The intestinal villi were completely and neatly arranged, and no inflammatory cell infiltration was observed in the propria nor edema in the saline group. We observed that the thickness of the mucosa increased slightly, the membrane propria was loosened, there were a few inflammatory cells, and only a small part of the intestinal villi was damaged in the control group. Typical pathological features of intestinal allergy were observed in the intestinal tissues of allergic mice, such as a large number of ruptured intestinal villi, inflammatory cell infiltration, and intestinal mucosal edema. Therefore, the β-CG allergy was determined to have caused cell apoptosis of the intestinal tissue based on the number of apoptotic cells and the pathological status of the intestinal tissue.

### 3.2. LGG Reduces Intestinal Apoptotic Cells Induced by β-CG Allergy

TUNEL assay was used to detect the number of apoptotic cells in the duodenum, jejunum, ileum, and colon (Figure 2A). The number of apoptotic cells were significantly reduced in the LGG-1 and LGG-2 groups. Compared with the NR group, the decrease in the percentage of apoptotic cells was more significant in the LGG group. The results indicated that LGG could effectively alleviate cell apoptosis induced by β-CG allergy, and the effect was better than that of NR.

The TUNEL assay was also used to determine the pathological status of the different parts of the intestinal tissue, including the duodenum, jejunum, ileum, and colon (Figure 2B–E). In the LGG group, there was no massive shedding of intestinal villi. The arrangement of intestinal villi and the thickness of the mucosal muscle layer returned to normal levels. The same pattern of apoptosis was observed in other parts of the intestine. The results showed that LGG intervention reduced the number of apoptotic cells and restored intestinal damage.

### 3.3. LGG Regulates the Imbalance of Gut Microbiota Induced by β-CG Allergy

The 16S rRNA sequencing analysis revealed the composition of gut microbiota in the intestine at the genus level (Figure 3A). We found that in allergic mice, the abundance of *Bacteroides* decreased while *Cerasibacillus* and unidentified *_Lachnospiraceae* increased. By contrast, the abundance of *Bacteroides* increased, and *Cerasibacillus* and unidentified *_Lachnospiraceae* decreased in the LGG group. The heat map was clustered through levels of species and samples (Figure 3B) and showed that the abundance of *Acidothermus*, *Cerasibacillus*, *Roseburia,* and unidentified *_Lachnospiraceae* increased after stimulation with β-CG. After LGG intervention, the abundance of *Roseburia* and unidentified *_Lachnospiraceae* decreased. In summary, these results indicated that β-CG allergy induced imbalance in gut microbiota and that LGG intervention could regulate this imbalance.

### 3.4. LGG Alleviates Intestinal Metabolites Disorder Induced by β-CG Allergy

The metabolic profiles of the intestinal contents were analyzed by LC-MS/MS in positive and negative modes (Figure 4A–D). The clear separation between the LGG-1 and NR-1, and the LGG-2 and β-CG groups was obtained by Partial Least Squares Discriminant Analysis (PLS-DA), indicating that LGG intervention significantly affected metabolites. The LGG-1 and NR-1 groups were compared in positive (R^2^ = 0.99 and Q^2^ = 0.91) and negative (R^2^ = 0.99 and Q^2^ = 0.91) ion modes. The LGG-2 and β-CG groups were compared, with R^2^ = 0.99 and Q^2^ = 0.64 in positive ion mode, and R^2^ = 0.99 and Q^2^ = 0.54 in negative ion mode. The results showed that the model had good interpretation and prediction ratios.

The sorting test demonstrated the correlations between the LGG-1 group and the NR-1 group, and between the LGG-2 group and the β-CG group, which were R^2^ (0.0, 0.88) and Q^2^ (0.0, −0.78) in positive ion mode, R^2^ (0.0, 0.88) and Q^2^ (0.0, −0.72) in negative ion mode, and R^2^ (0.0, 0.97) and Q^2^ (0.0, 0.67) in positive ion mode, and R^2^ (0.0, 0.97) and Q^2^ (0.0, 0.61) in negative ion mode, respectively (Figure 4E–H). The results showed that the model was not over-fitted and, thus, could describe the sample well and be used in the search for differential metabolite.

Hierarchical clustering analysis was performed on the differential metabolites in which the same or similar metabolic patterns were clustered and used to infer the function of unknown or known metabolites. Compared with the NR-1 group, we found obvious metabolic differences in positive ion mode in the LGG-1 group (Figure 5A). Figure 5B showed the comparison of differential metabolites of the LGG-2 and β-CG groups in positive ion mode. Note areas with different colors, representing different clustering information; the results revealed that the mice subject to LGG intervention and β-CG allergic mice used different metabolic pathways and, hence, had different metabolites in their intestine.

Figure 5C,D shows the correlation graph of differential metabolites in positive ion mode. Differential metabolites have a synergistic or mutually exclusive relationship. We analyzed the correlation between each metabolite by calculating Pearson correlation coefficients between all metabolites. When the linear relationship between the two metabolites increases, the positive correlation tends to 1, and the negative correlation tends to −1. The above results revealed that when comparing the LGG-1 group with the NR-1 group, Com_10093_pos was positively correlated with Com_10177_pos, Com_10402_pos, Com_10750_pos, and Com_10008_pos; Com_10297_pos was positively correlated with Com_10240_pos; Com_1007_pos was negatively correlated with Com_10001_pos; Com_10428_pos was negatively correlated with Com_10093_pos and Com_10177_pos. In comparing the LGG-2 and β-CG groups, Com_11020_pos was positively correlated with Com_12092_pos; Com_11284_pos was positively correlated with Com_10082_pos; Com_11102_pos was positively correlated with Com_12252_pos and negatively correlated with Com_10116_pos and Com_11859_pos.

The Z-score is a transformed value based on the relative content of differential metabolites in the mouse intestine and is used to compare the similarity of the differential metabolites. Figure 5E,F showed the relative metabolite contents of the first 30 compounds in the LGG-1, NR-1, LGG-2, and β-CG groups. The Z-score analysis revealed the samples had good repeatability.

According to the above results, a bubble chart of the enriched Kyoto Encyclopedia of Genes and Genomes (KEGG) pathway was drawn. Compared with the NR-1 group, the main metabolic pathways in the LGG-1 group were the bile secretion pathway, histamine metabolism, and glycerophospholipid metabolism (Figure 6A). Compared with the β-CG group, the main metabolic pathway in the LGG-2 group was bile secretion pathway (Figure 6B). The results showed that the most differentially abundant metabolites after LGG intervention were of the bile secretion pathway.

Compared with the NR-1 group, histamine and *N*(α)-γ-glutamylhistamine were significantly downregulated in the LGG-1 group (Figure 7A). The results showed that allergic biomarkers were downregulated in the LGG group more than in the NR group, indicating that LGG intervention was more effective. Compared with the β-CG allergic mice (Figure 7B), histamine, *N*(α)-γ-glutamylhistamine, *N*-acetylhistamine, arachidonic acid, phenylalanine, tryptophan, malate, and xanthine were significantly downregulated in the LGG-2 group, whereas deoxycholic acid and lithocholic acid were significantly upregulated. The results showed that after LGG intervention, the levels of allergic biomarkers were significantly lowered.

## 4. Discussion

Apoptosis induced by allergies has recently received increased attention. Usually, the allergic response leads to intestinal damage, such as the rupture of intestinal villi, increased intestinal permeability, and increased numbers of apoptotic cells. In our previous work [22], we found that LGG alleviates β-CG allergy by modulating the differentiation of immune cells in the TCR signaling pathway. However, the effect of LGG on alleviating cell apoptosis has not been reported. LGG is recognized to regulate the gut microbiota and metabolites; therefore, in this study, we explored the effects of LGG intervention on cell apoptosis induced by β-CG allergy and the related mechanisms and metabolites in gut microbiota.

Our experiments demonstrated that β-CG allergy induced cell apoptosis, including the occurrence of intestinal inflammation and the destruction of the intestinal wall barrier. Following LGG intervention, apoptotic cells were significantly decreased, and the state of intestinal villi was restored. It has been reported that the development of normal tolerance of apoptotic cells is related to pro-inflammatory bacteria [24,25]. Probiotics can reduce pro-inflammatory bacteria, and maintain intestinal immune homeostasis [26]. Therefore, LGG intervention may reduce the number of apoptotic cells by lowering the tolerance of apoptotic cells.

Probiotic-related immunomodulatory effects on the intestinal tract have been verified and reported. It has been reported that probiotic *Enterococcus faecium* has a regulatory effect on necrotic enteritis-induced intestinal barrier damage [27]. Probiotic *Pediococcus acidilactici* can restore intestinal morphology and enteric immunity [28]. LGG has been demonstrated to possess the recovery effect on intestinal inflammation induced by lipopolysaccharide [29]. It was observed that β-CG-induced allergy could lead to an increase in the number of apoptotic cells and rupture of intestinal villi in current study, whereas LGG intervention was verified to have the functionality to effectively reduce the number of apoptotic cells and restore the normal state of intestinal villi. The underlying mechanism may rely on the enhanced intestinal mucosal barrier function and intestinal immune response induced by LGG, which facilitated the recovery of the intestinal tissue damage. LGG has the capacity to inhibit cell apoptosis of the intestine, regulate homeostasis of intestinal epithelium, and prevent intestinal inflammatory diseases. On this basis, further research revealed that LGG’s inhibition of cell apoptosis is related to the gut microbiota. The gut microbiota in healthy individuals is in a dynamic equilibrium state, whereas allergic mice were in an imbalanced state. From our experimental results, the relative abundance of *Enterobacter* increased in β-CG allergic mice and then decreased after LGG intervention. According to reports, the imbalance of gut microbiota is directly related to allergies [30]. The microbial composition in the intestinal tract of normal mice is diverse, and the proportion of *Enterobacter* is low. Bridgman et al. reported changes in the gut microbiota induced by children’s allergies, in which the abundance of *Enterobacter* increased [31]. Thus, β-CG allergy induced an increase in *Enterobacter*, and LGG intervention reduced *Enterobacter*. The results of this study have showed that decreased *Bacteroides* in the intestine of allergic mice, which increased after LGG intervention. *Bacteroides* is the most common beneficial bacteria in the human ileum and large intestine, and the main component of the host’s normal flora. *Bacteroides* has a symbiotic relationship with the host. Some studies have confirmed that intestinal bacteria, such as *Bifidobacteria* and *Lactobacilli*, are involved in reducing the risk of allergic diseases. *Enterococcus* and other allergic bacteria, such as *Enterobacteriaceae* and *Clostridia*, increase the risk of developing allergies [32]. Reddel et al. showed that probiotics promote the abundance of *Bacteroides* in the intestine of infants [33]. Therefore, the results revealed that LGG could restore normal microbiota, reduce pro-inflammatory bacteria, and regulate the imbalance in the gut microbiota.

The metabolic pathways of the gut microbiota are essential for maintaining intestinal immune homeostasis, and they affect the host’s immunity to a variety of immune-mediated diseases. In the metabolic results, the bile secretion pathway was significantly enriched after LGG intervention. The metabolites produced by the bile secretion pathway have immunomodulatory functions [34]. Thus, we infer that LGG intervention could play an immunomodulatory role in the bile secretion pathway.

Future research needs to further identify the key metabolites in the bile secretion pathway because the mutual regulation between the host and its microbiome occurs through the secretion of metabolites. The bile secretion pathway can be achieved by regulating the metabolic products of the metabolic and inflammatory pathways. During β-CG allergy, histamine is present in relatively high concentrations in the bile secretion pathway. After LGG intervention, key differential metabolites regulate the immune responses, including upregulation of deoxycholic acid and cholic acid and downregulation of histamine in the bile secretion pathway. Recent studies have reported that metabolites play an essential role in the immune system [35]. Histamine is a biogenic amine that has a wide range of effects on many cell types, and the activation of receptors (H1R-H4R) mediates this effect, such as causing metabolic abnormalities and allergic diseases. The concentration of histamine depends on the expression and activity of histamine receptors [36]. Histamine metabolism is closely related to food allergy [37]. Deoxycholic acid and lithocholic acid play a conductive role in maintaining the homeostasis of the bile secretion pathway. Deoxycholic acid and lithocholic acid can activate FXR, after which FXR heterodimerizes with 9-cis retinoic X receptor (RXR) to regulate cholesterol catabolism and BA biosynthesis [38]. Therefore, LGG intervention could regulate key differential metabolites in the bile secretion pathway.

The gut microbiota and its metabolites are closely related, and their interaction plays an important role in the immune system. *Bacteroides* express bile salt hydrolase (BSH), which deconjugates taurine-conjugated BAs and glycine-conjugated BAs. Metabolites may activate FXR in the intestine to regulate the bile secretion pathway and participate in the immune regulatory system. The role of the bile secretion pathway in the immune system is to increase the expression of *Muc2* in the intestine and induce production of the components of mucosal immune cells, thereby reducing apoptotic cells. In conclusion, we think that apoptotic cells in the intestine were decreased due to the increase in *Bacteroides,* which promoted the bile secretion pathway following LGG intervention (Figure 8).

## 5. Conclusions

In this study, we found through TUNEL analysis that LGG could reduce cell apoptosis induced by β-CG allergy. Further analysis of the gut microbiota and metabolic pathways of the intestinal contents revealed that LGG intervention regulated *Bacteroides* levels and the bile secretion pathway. As a new perspective for reducing apoptosis of cells, our findings provide a basis for the application of dietary intervention in the treatment of allergies.

## Figures and Tables

**Figure 1 nutrients-13-00055-f001:**
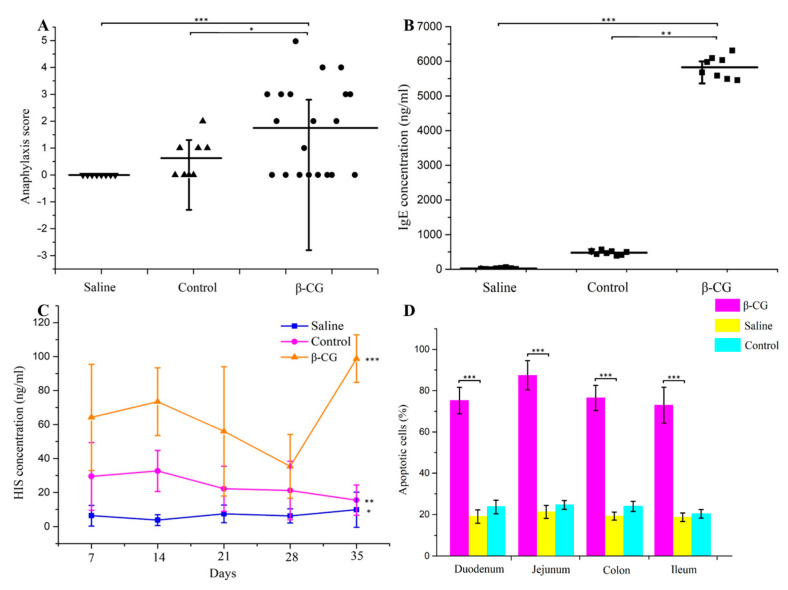
β-conglycinin (β-CG) induced intestinal cell apoptosis in allergy mouse. (**A**) The score of anaphylaxis symptoms; (**B**) Levels of IgE; (**C**) Levels of histamine (HIS). (**D**) Terminal deoxynucleotidyl transferase-mediated dUTP nick-end labeling (TUNEL) assay on the number of apoptotic cells in duodenal, jejunum, ileum, colon. TUNEL assay on the pathological status in (**E**) duodenal; (**F**) jejunum; (**G**) ileum; (**H**) colon. (* *p* < 0.05; ** *p* < 0.01; *** *p* < 0.001).

**Figure 2 nutrients-13-00055-f002:**
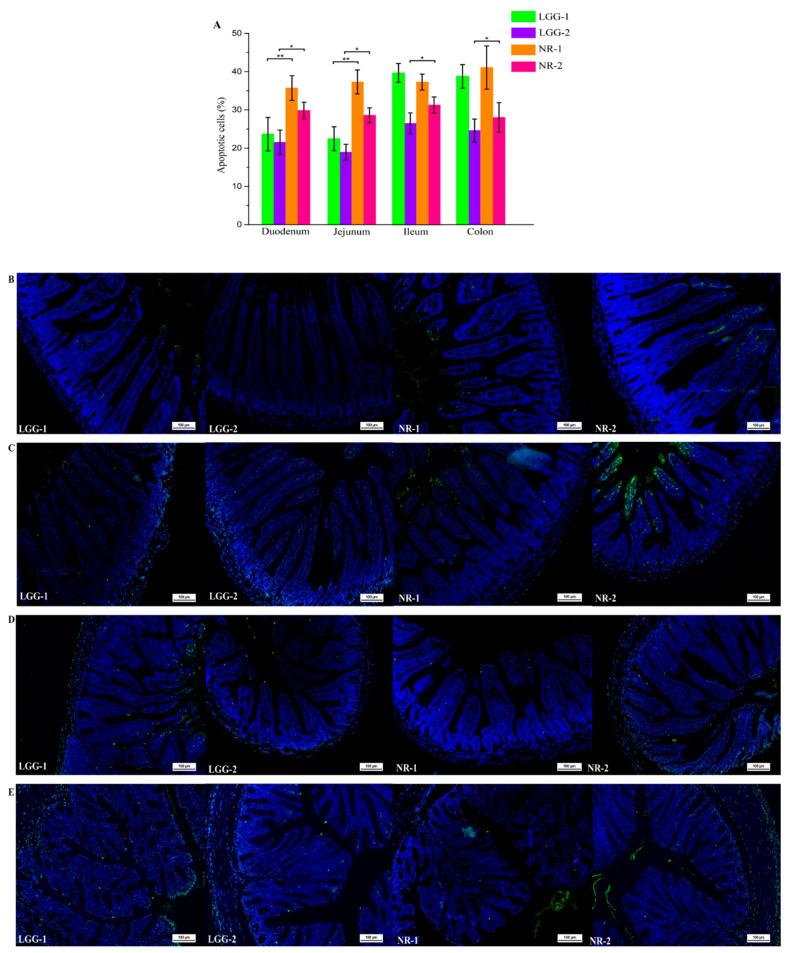
(**A**) TUNEL assay on the number of apoptotic cells in duodenal, jejunum, ileum and colon. TUNEL assay on the pathological status in (**B**) duodenal; (**C**) jejunum; (**D**) ileum; (**E**) colon. (* *p* < 0.05; ** *p* < 0.01).

**Figure 3 nutrients-13-00055-f003:**
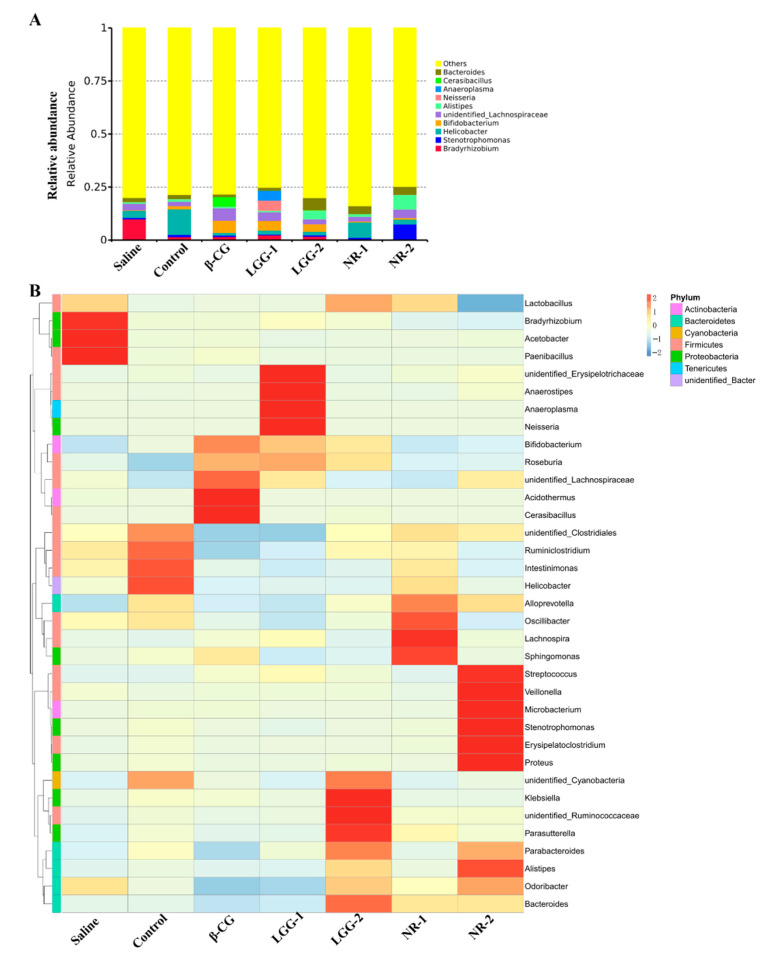
The genus level analysis of gut microbiota in intestinal contents. (**A**) Histogram of relative abundance of gut microbiota; (**B**) Heat map of gut microbiota.

**Figure 4 nutrients-13-00055-f004:**
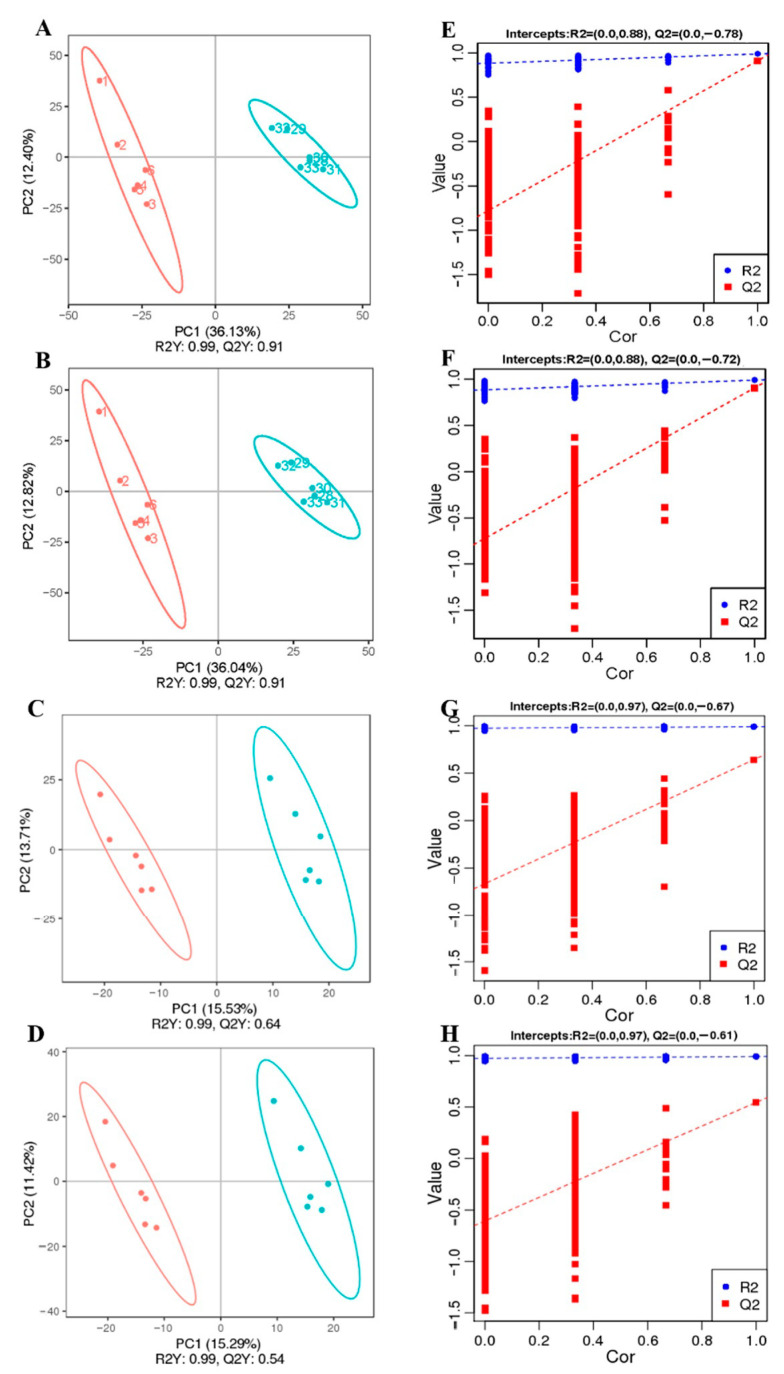
Partial Least Squares Discriminant Analysis (PLS-DA) analysis of intestinal contents. (**A**) PLS-DA scores plots depicting obvious difference between the *Lactobacillus rhamnosus* GG (LGG)-1 group and the natural recovery (NR)-1 group in positive ion mode. (**B**) PLS-DA scores plots depicting obvious difference between the LGG-1 group and the NR-1 group in negative ion mode. (**C**) PLS-DA scores plots depicting obvious difference between the LGG-2 group and the β-CG group in positive ion mode. (**D**) PLS-DA scores plots depicting obvious difference between the LGG-2 group and the β-CG group in negative ion mode. (**E**) Sorting verification chart depicting obvious difference between the LGG-1 group and the NR-1 group in positive ion mode. (**F**) Sorting verification chart depicting obvious difference between the LGG-1 group and the NR-1 group in negative ion mode. (**G**) Sorting verification chart depicting obvious difference between the LGG-2 group and the β-CG group in positive ion mode. (**H**) Sorting verification chart depicting obvious difference between the LGG-2 group and the β-CG group in negative ion mode.

**Figure 5 nutrients-13-00055-f005:**
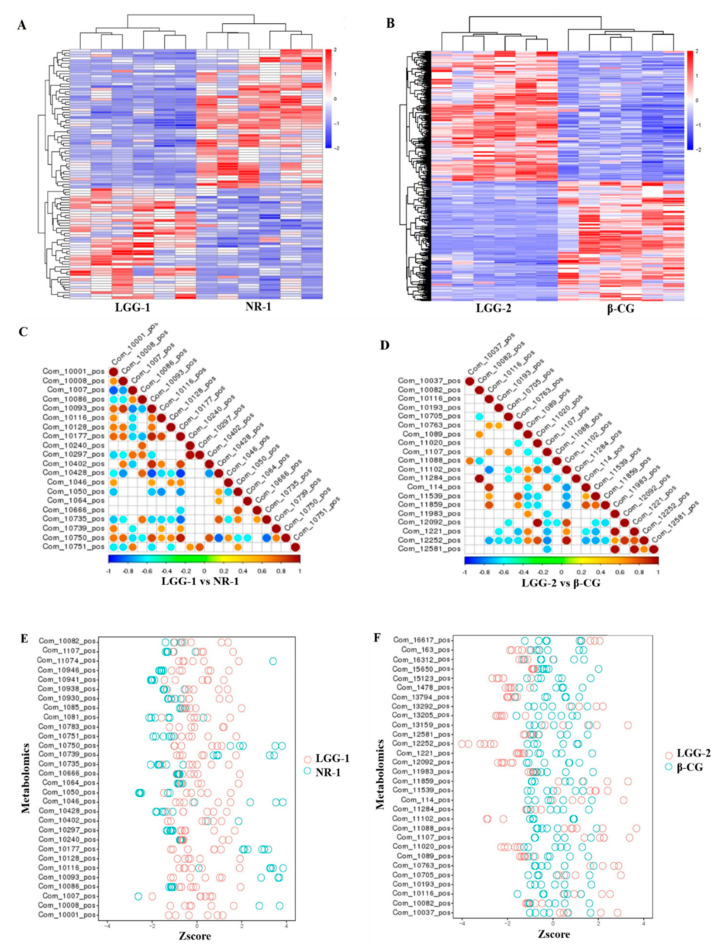
Analysis of differential metabolites between different groups. (**A**) Cluster heat map of differential metabolites in positive ion mode between the LGG-1 group and the NR-1 group. (**B**) Cluster heat map of differential metabolites in positive ion mode between the LGG-2 group and the β-CG group; (**C**) Correlation diagram of differential metabolites in positive ion mode between the LGG-1 group and the NR-1 group; (**D**) Correlation diagram of differential metabolites in positive ion mode between the LGG-2 group and the β-CG group; (**E**) Z-score chart in positive ion mode between the LGG-1 group and the NR-1 group; (**F**) Z-score chart in positive ion mode between the LGG-2 group and the β-CG group.

**Figure 6 nutrients-13-00055-f006:**
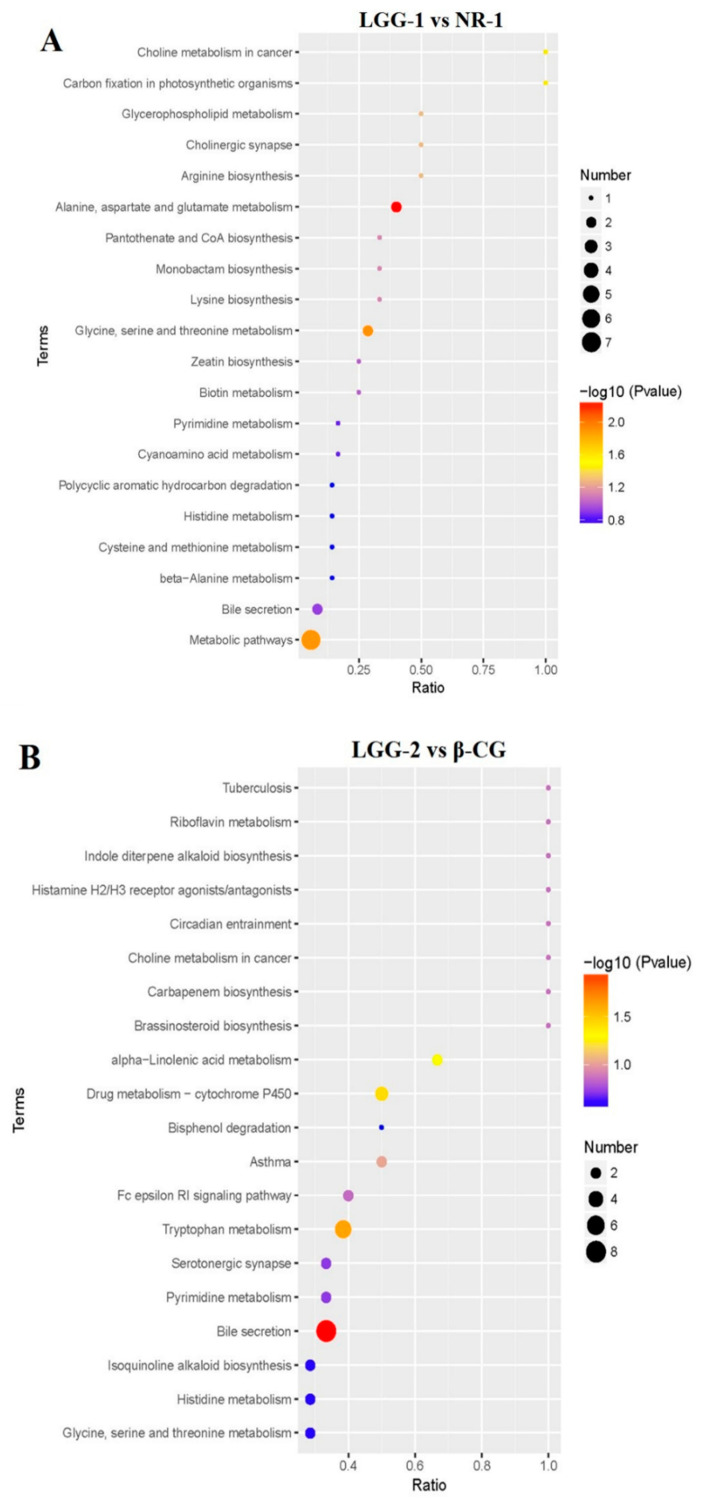
Comparing the bubble chart of the enriched KEGG pathway between different groups. (**A**) Metabolomic pathway chart in positive ion mode between the LGG-1 group and the NR-1 group; (**B**) Metabolomic pathway chart in positive ion mode between the LGG-2 group and the β-CG group.

**Figure 7 nutrients-13-00055-f007:**
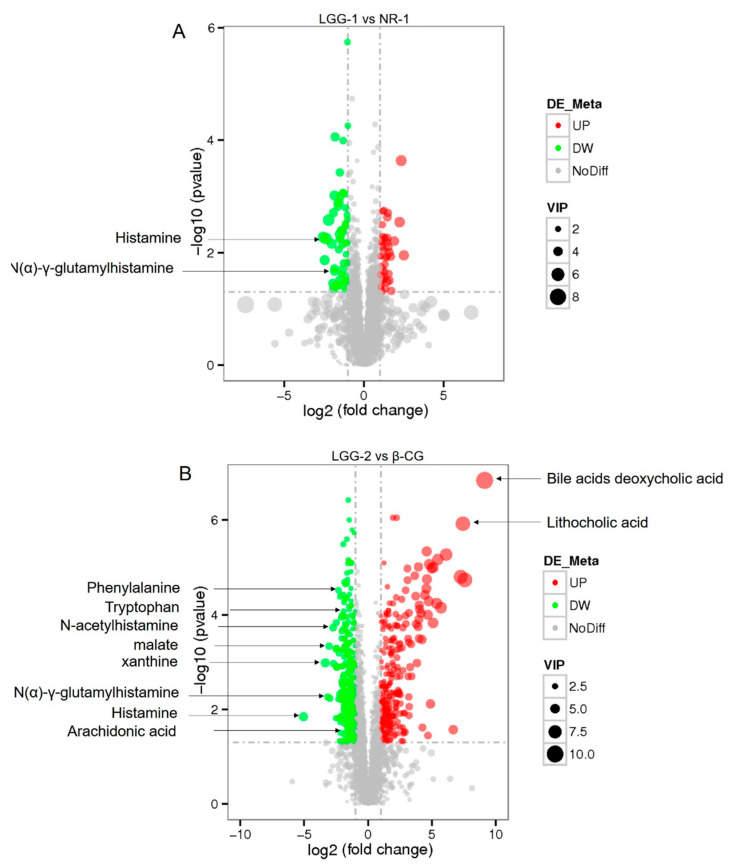
Identification of metabolites in the bile secretion pathway. (**A**) Volcano map of differential metabolites in positive ion mode between the LGG-1 group and the NR-1 group; (**B**) Volcano map of differential metabolites in positive ion mode between the LGG-2 group and the β-CG group.

**Figure 8 nutrients-13-00055-f008:**
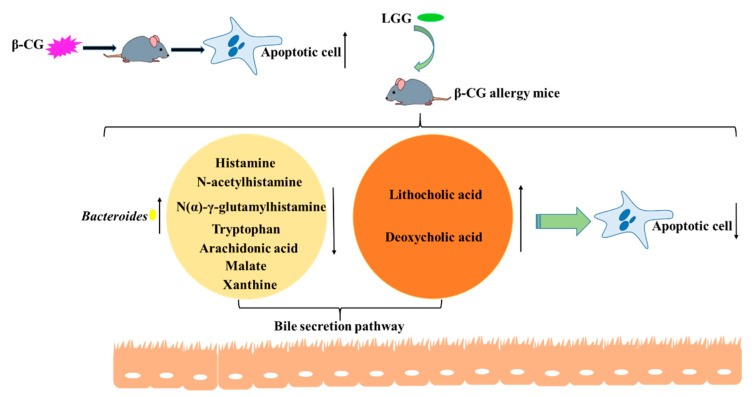
Mechanism of LGG reducing apoptotic cells through regulating *Bacteroides* levels and the bile secretion pathway.

## Data Availability

The data presented in this study are available on request from the corresponding author. The data are not publicly available due to ethical and privacy reasons.

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
