# Peer review of "Lactobacillus rhamnosus GG Reduces β-conglycinin-Allergy-Induced Apoptotic Cells by Regulating Bacteroides and Bile Secretion Pathway in Intestinal Contents of BALB/c Mice"

_nutrients, 2020, doi:10.3390/nu13010055_

Round 1
Reviewer 1 Report
The manuscript by Xiaoxu and colleagues described that LGG could reduce cell apoptosis that was induced by β-CG allergy. The analyses of the gut microbiota and metabolic pathways revealed that LGG intervention was mediated via regulating Bacteroides and bile secretion pathway in intestinal contents. This is a very interesting, well-conducted study. It provides a novel insight into treating allergies through dietary intervention. Please modify the texts in these figures because they are too tiny to be read. Also, some typos and grammatical errors should be double-checked and fixed in this manuscript.
Author Response
Responds to comments
The revised content in the updated manuscript has been highlighted with red font. The detailed response to the comments from reviewers has been listed below in the following section.
Responds to Reviewer #1:
- Please modify the texts in these figures because they are too tiny to be read.
Response: Great appreciation for your suggestion. The texts in the related figures have been updated in Figure 1A-H, Figure 2A-E, Figure 3A-B, Figure 5A-F, Figure 6A-B, and Figure 7A-B in the revised manuscript.
- Also, some typos and grammatical errors should be double-checked and fixed in this manuscript.
Response: The English language, grammar, punctuation, spelling, and overall style have been polished and revised throughout the article by a highly qualified English-speaking editor provided by MDPI. The language has been double checked, please review the section highlighted in red font in the revised manuscript.

Reviewer 2 Report
Summary:
This study investigates a potential underlying mechanism of Lactobacillus rhamnosus GG(LGG) in reducing beta-conglycinin(b-CG) induced apoptosis of intestinal cells in mouse model system. They demonstrate that, b-CG potentially induces intestinal apoptosis in allergic mice and which is rescued by LGG. This could be due to the alleviation of intestinal metabolites induced by b-CG. Their findings suggest that LGG intervention regulated bacteroids and bile secretion pathways which in turn provide a good insight for the application of dietary intervention in the treatment of allergies.
Overall:
The experiments are very well done and the manuscript is well written. The data is novel, clear and easy to interpret. I have just few important concerns that need to be clarified.
- Authors present a very clean data of b-CG inducing apoptosis, later LGG intervention reducing the apoptosis and restoration of intestinal villi. What exactly authors are trying to present here? The role of LGG is bit confusing, is LGG playing duel role? As per the data it is clear that apoptotic + cells were decreased in LGG treated group, which suggest role of LGG in inhibiting apoptosis. Also authors present data of restoration of intestinal villi, will this suggest the reparative/regenerative role of LGG? Discuss this and be clear to not to make readers confuse (Discussion paragraph 3 is not sufficiently suggesting this).
- What would be the fate of intestinal cells in LGG alone received animals without prior supplementation with b-CG?
Minor:
Overall methodological aspect of data analysis of Sr RNA seq sections needs to be elaborated as readers may cite this paper for the appropriate methodological use if they find relevant.
How did the genome alignment done? What are the packages used for genome alignment.
How did the downstream analysis and visualization of the data performed? Which platform used for downstream analysis (R/Python)?
How did differential metabolite analysis performed?
How does KEGG pathway analysis performed (R/Python based packages or online data base)?
Below mentioned figures axis legends were irritating for eyes and need to be changed as readable.
Fig3B, Fig 5E and F, and the key data of paper Fig6A and B.
Author Response
Responds to comments
The revised content in the updated manuscript has been highlighted with red font. The detailed response to the comments from reviewers has been listed below in the following section.
Responds to Reviewer #2:
- Authors present a very clean data of β-CG inducing apoptosis, later LGG intervention reducing the apoptosis and restoration of intestinal villi. What exactly authors are trying to present here? The role of LGG is bit confusing, is LGG playing duel role? As per the data it is clear that apoptotic cells were decreased in LGG treated group, which suggest role of LGG in inhibiting apoptosis. Also authors present data of restoration of intestinal villi, will this suggest the reparative/regenerative role of LGG? Discuss this and be clear to not to make readers confuse (Discussion paragraph 3 is not sufficiently suggesting this).
Response: Many thanks for the critical comments. Up to date, there is no experimental elucidation about the intervention effect of LGG on apoptotic cells and intestinal villi. In current research, it was demonstrated that LGG can reduce cell apoptosis and restore intestinal villi. Herein, the dual role of LGG could be summarized. LGG can reduce apoptotic cells and restore intestinal villi by regulating the level of Bacteroides and through bile acid secretion pathways. Meanwhile, supplement of the detailed discussion in paragraph 3 has been provided in the revised manuscript.
- What would be the fate of intestinal cells in LGG alone received animals without prior supplementation with β-CG?
Response: Many thanks for your comments. Because the effect of LGG on intestinal cells has been reported in previous researches, including LGG can regulate the intestinal immune response, reduce pro-inflammatory cytokines, protect the function of intestinal epithelial cells, maintain intestinal cell homeostasis, so we did not gavage mice with only LGG to analyze the effect on intestinal cells.
DOI: https://doi.org/10.1038/mi.2016.43;
DOI: https://doi.org/10.1038/s41598-017-13466-1;
DOI: https://doi.org/10.1016/j.colsurfb.2018.03.044.
- How did the genome alignment done? What are the packages used for genome alignment.
Response: Sequences analysis were performed by using Uparse software (Uparse v7.0.1001, http://drive5.com/uparse/). Sequences with ≥ 97% similarity were assigned to the same OTUs. Representative sequence for each OTU was screened for further annotation. For each representative sequence, the Silva Database (https://www.arb-silva.de/) was employed to annotate taxonomic information based on Mothur algorithm. Multiple sequence alignment were conducted by using the MUSCLE software (Version 3.8.31, http://www.drive5.com/muscle/) to study phylogenetic relationship of different OTUs, as well as the difference of the dominant species in different samples (groups). Related information has been included in the section of the materials and methods (2.8 16S rRNA Gene Sequencing Analysis) in the revised manuscript.
- How did the downstream analysis and visualization of the data performed? Which platform used for downstream analysis (R/Python)?
Response: The related downstream analysis and data visualization were accomplished with QIIME (Version1.7.0) and displayed with R software (Version 2.15.3). Related information has been supplemented to the materials and methods (2.8 16S rRNA Gene Sequencing Analysis) in the revised manuscript.
- How did differential metabolite analysis performed?
Response: The raw data files generated after UHPLC-MS/MS were processed by using the Compound Discoverer 3.0 (CD 3.0, Thermo Fisher) to perform peak alignment, peak picking, and quantitation for each metabolite. The main parameters were set as following: retention time tolerance, 0.2 minutes; actual mass tolerance, 5ppm; signal intensity tolerance, 30%; signal/noise ratio, 3; and minimum intensity,100000. After that, peak intensities were normalized to the total spectral intensity. The normalized data was used to predict the molecular formula based on additive ions, molecular ion peaks and fragment ions. And then peaks were matched with the mzCloud (https://www.mzcloud.org/) and ChemSpider (http://www.chemspider.com/) database to obtained the accurate qualitative and relative quantitative results. Related information has been added to the part of materials and methods (2.9 Metabolomics Analysis) in the revised manuscript.
- How does KEGG pathway analysis performed (R/Python based packages or online data base)?
Response: KEGG pathway analysis was packaged based on Python software (Python-3.5.0). Related information has been included in materials and methods (2.9 Metabolomics Analysis) in the revised manuscript.
- Below mentioned figures axis legends were irritating for eyes and need to be changed as readable. Fig3B, Fig 5E and F, and the key data of paper Fig6A and B.
Response: Many thanks for the suggestions. The Figures have been rescaled to be visualized as Figure 3B, Figure 5E-F, and Figure 6A-B in the revised manuscript.

Reviewer 3 Report
Drs Chen et all present an interesting study on probiotic supplementation for the prevention of allergy -induced intestinal cell apoptosis and identify bile acid signalling pathways as a possible link in this mechanism of probiotic action.
The study has sound methodology.
The analysis has been described well.
The discussion is clear and balanced.
minor comments regarding style:
section 2.7: .., and we observation - sentecne needs restructuring
discussion page 12: suggest `healthy individuals` instead of ` normal people`
Author Response
The revised content in the updated manuscript has been highlighted with red font. The detailed response to the comments from reviewers has been listed below in the following section.
Responds to Reviewer #3:
- section 2.7, and we observation - sentence needs restructuring.
Response: Many thanks for your comments. This sentence has been reorganized and double-checked. Please see the sentence highlighted in red font in section 2.7 in the revised manuscript.
- discussion page 12: suggest “healthy individuals” instead of “normal people”.
Response: Many thanks for your suggestion. The description of “normal people” has been updated with “healthy individuals”. Please check the highlighted revised sentence in red font in discussion of the revised manuscript.
